# Generative Noisy-Label Learning by Implicit Dicriminative Approximation with Partial Label Prior

## Abstract

The learning with noisy labels has been addressed with both discriminative and generative models. Although discriminative models have dominated the field due to their simpler modeling and more efficient computational training processes, generative models offer a more effective means of disentangling clean and noisy labels and improving the estimation of the label transition matrix. However, generative approaches maximize the joint likelihood of noisy labels and data using a complex formulation that only indirectly optimizes the model of interest associating data and clean labels. Additionally, these approaches rely on generative models that are challenging to train and tend to use uninformative clean label priors. In this paper, we propose a new generative noisy-label learning approach that addresses these three issues. First, we propose a new model optimisation that directly associates data and clean labels. Second, the generative model is implicitly estimated using a discriminative model, eliminating the inefficient training of a generative model. Third, we propose a new informative label prior inspired by partial label learning as supervision signal for noisy label learning. Extensive experiments on several noisy-label benchmarks demonstrate that our generative model provides state-of-the-art results while maintaining a similar computational complexity as discriminative models. *Code will be available once paper is accepted.*

## 1   Introduction

Deep neural network (DNN) has achieved remarkable success in computer vision [13, 21], natural language processing (NLP) [10, 51] and medical image analysis [24, 38]. However, DNNs often require massive amount of high-quality annotated data for supervised training [9], which is challenging and expensive to acquire. To alleviate such problem, some datasets have been annotated via crowdsourcing [46], from search engines [35], or with NLP from radiology reports [38]. Although these cheaper annotation processes enable the construction of large-scale datasets, they also introduce noisy labels for model training, resulting in performance degradation. Therefore, novel learning algorithms are required to robustly train DNN models when training sets contain noisy labels.

The main challenge in noisy-label learning is that we only observe the data, represented by random variable $X$, and respective noisy label, denoted by variable $\tilde{Y}$, but we want to estimate the model $p(Y|X)$, where $Y$ is the hidden clean label variable. Most methods proposed in the field resort to two discriminative learning strategies: sample selection and noise transition matrix. *Sample selection* [1, 12, 22] optimises the model $p_\theta(Y|X)$, parameterised by $\theta$, with maximum likelihood optimisation restricted to pseudo-clean training samples, as follows

$$\theta^* = \arg\max_\theta \mathbb{E}_{P(X,\tilde{Y})} \left[ \mathsf{clean}(X, \tilde{Y}) \times p_\theta(\tilde{Y}|X) \right], \text{ where } \mathsf{clean}(X = \mathbf{x}, \tilde{Y} = \tilde{\mathbf{y}}) = \begin{cases} 1, \text{ if } Y = \tilde{\mathbf{y}} \\ 0, \text{ otherwise} \end{cases},$$

$$(1)$$

and $P(X, \tilde{Y})$ is the distribution used to generate the noisy-label and data points for the training set. Note that $\mathbb{E}_{P(X,\tilde{Y})} \left[ \mathsf{clean}(X, \tilde{Y}) \times p_\theta(\tilde{Y}|X) \right] \equiv \mathbb{E}_{P(X,Y)} \left[ p_\theta(Y|X) \right]$ if the function $\mathsf{clean}(.)$ successfully selects the clean-label training samples. Unfortunately, $\mathsf{clean}(.)$ usually relies on the *small-loss hypothesis* [2] for selecting $R\%$ of the smallest loss training samples, which offers little guarantees of successfully selecting clean-label samples. Approaches based on noise *transition matrix* [44, 6, 32] aim to estimate a clean-label classifier and a label transition, as follows:

$$\theta^* = \arg\max_\theta \mathbb{E}_{P(X,\tilde{Y})} \left[ \sum_Y p(\tilde{Y}|X) \right] = \arg\max_{\theta_1,\theta_2} \mathbb{E}_{P(X,\tilde{Y})} \left[ \sum_Y p_{\theta_1}(\tilde{Y}|Y,X) p_{\theta_2}(Y|X) \right], \quad (2)$$

where $\theta = [\theta_1, \theta_2]$, $p_{\theta_1}(\tilde{Y}|Y, X)$ represents a label-transition matrix, often simplified to be class-independent with $p_{\theta_1}(\tilde{Y}|Y) = p_{\theta_1}(\tilde{Y}|Y, X)$. Since we do not have access to the label transition matrix, we need to estimate it from the noisy-label training set, which is challenging because of identifiability issues [27], making necessary the use of anchor point [32] and regularisations [6].

On the other hand, generative learning models [3, 11, 50] assume a generative process for $X$ and $Y$, as described in Fig. 1. These methods are trained to maximise the data likelihood $p(\tilde{Y}, X) = \int_{Y,Z} p(X|Y, Z) p(\tilde{Y}|Y, X) p(Y) p(Z) dY dZ$, where $Z$ denotes a latent variable representing a low-dimensional representation of the image, and $Y$ is the latent clean label. This optimisation requires a variational distribution $q_\phi(Y, Z|X)$ to maximise the evidence lower bound (ELBO): with

$$\theta_1^*, \theta_2^*, \phi^* = \arg\max_{\theta_1,\theta_2,\phi} \mathbb{E}_{q_\phi(Y,Z|X)} \left[ \log \left( p_{\theta_1}(X|Y, Z) p_{\theta_2}(\tilde{Y}|X, Y) p(Y) p(Z) / q_\phi(Y, Z|X) \right) \right], \quad (3)$$

where $p_{\theta_1}(X|Y, Z)$ denotes an image generative model, $p_{\theta_2}(\tilde{Y}|X, Y)$ represents the label transition model, $p(Z)$ is the latent image representation prior (commonly assumed to a standard normal distribution), and $p(Y)$ is the clean label prior (usually assumed to be a non-informative prior based on a uniform distribution). Such generative strategy is sensible because it disentangles the true and noisy labels and improves the estimation of the label transition model [50]. A limitation of the generative strategy is that it optimises $p(\tilde{Y}, X)$ instead of directly optimising $p(X|Y)$ or $p(Y|X)$. Also, compared with the discriminative strategy, the generative approach requires the generative model $p_{\theta_1}(X|Y, Z)$ that is challenging to train. This motivates us to ask the following question: **Can we directly optimise the generative goal $p(X|Y)$, with a similar computational cost as the discriminative strategy and accounting for an informative prior for the latent clean label $Y$?**

In this paper, we propose a new generative noisy-label learning method to directly optimise $p(X|Y)$ by maximising $\mathbb{E}_{q(Y|X)} \left[ \log p(X|Y) \right]$ using a variational posterior distribution $q(Y|X)$. This objective function is decomposed into three terms: a label-transition model $\mathbb{E}_{q(\mathbf{y}|\mathbf{x})} \left[ \log p(\tilde{\mathbf{y}}|\mathbf{x}, \mathbf{y}) \right]$, an image generative model $\mathbb{E}_{q(\mathbf{y}|\mathbf{x})} \left[ \log \frac{p(\mathbf{x}|\mathbf{y})p(\mathbf{y})}{q(\mathbf{y}|\mathbf{x})} \right]$, and a Kullback–Leibler (KL) divergence regularisation term. We implicitly estimate the image generative term with the discriminative model $q(Y|X)$, bypassing the need to train a generative model. Moreover, our formulation allows the introduction of an instance-wise informative prior $p(Y)$ inspired by partial-label learning [36]. This prior is re-estimated at each training epoch to cover a small number of label candidates if the model is certain about the training label. Conversely, when the model is uncertain about the training label, then the label prior will cover a large number of label candidates, which also serve as a regularisation of noisy label training. Our formulation only requires a discriminative model and a label transition model, making it computationally less expensive than other generative approaches [3, 11, 50]. Overall, our contributions can be summarized as follows:

- We introduce a new generative framework to handle noisy-label learning by directly optimising $p(X|Y)$.

- Our generative model is implicitly estimated with a discriminative model, making it computationally more efficient than previous generative approaches [3, 11, 50].

- Our framework allows us to place an informative instance-wise prior $p(Y)$ for latent clean label $Y$. Inspired by partial label learning [36], $p(Y)$ is constructed for maintaining high coverage for latent clean label and regularise uncertain sample training.

We conduct extensive experiments on both synthetic and real-world noisy-label benchmarks that show that our method provides state-of-the-art (SOTA) results and enjoy a similar computational complexity as discriminative approaches.

## 2 Related Work

**Sample selection.** The discriminative learning strategy based on sample selection from (1) needs to handle two problems: 1) the definition of clean(.), and 2) what to do with the samples classified as noisy. Most definitions of clean(.) resort to classify small-loss samples [2] as pseudo-clean [1, 4, 12, 15, 22, 30, 34, 40]. Other approaches select clean samples based on the K nearest neighbor classification in an intermediate deep learning feature spaces [31, 39], distance to the class-specific eigenvector from the gram matrix eigen-decomposition using intermediate deep learning feature spaces [17], uncertainty measures [19], or prediction consistency between teacher and student models [16]. After sample classification, some methods will discard the noisy-label samples for training [4, 15, 30, 34], while others use them for semi-supervised learning [22]. The main issue with this strategy is that it does not try to disentangle the clean and noisy-label from the samples.

**Label transition model.** The discriminative learning strategy based on the label transition model from (2) depends on a reliable estimation of $p(\tilde{Y}|Y, X)$ [6, 32, 44]. Forward-T [32] uses an additional classifier and anchor points from clean-label samples to learn a class-dependent transition matrix. Part-T [44] estimates an instance-dependent model. MEDITM [6] uses manifold regularization for estimating the label-transition matrix. In general, the estimation of this label transition matrix is under-constrained, leading to the identifiability problem [27], which is addressed with the formulation of anchor point [32], or additional regularisation [6].

**Generative modelling.** Generative modeling for noisy-label learning [3, 11, 50] explores different graphical models (see Fig. 1) to enable the estimation of clean labels per image. Specifically, CausalNL [50] and InstanceGM [11] assume that the latent clean label $Y$ causes $X$, and the noisy label $\tilde{Y}$ is generated from $X$ and $Y$. Alternatively, NPC [3] assumes that $X$ causes $Y$ and proposes a post-processing calibration for noisy label learning. One drawback of generative modeling is that instead of directly optimising the models of interest $p(X|Y)$ or $p(Y|X)$, it optimises the joint distribution of visible variables $p(X, \tilde{Y})$. Even though maximising the likelihood of the visible data is sensible, it only produces the models of interest as a by-product of the process. Furthermore, these methods require the computationally complex training of a generative model, and usually rely on non-informative label priors.

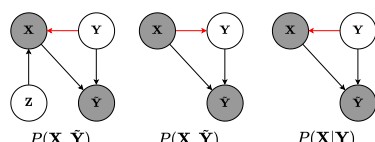

$P(\mathbf{X}, \tilde{\mathbf{Y}})$ $\quad$ $P(\mathbf{X}, \tilde{\mathbf{Y}})$ $\quad$ $P(\mathbf{X}|\mathbf{Y})$

Figure 1: Generative noisy-label learning models and their corresponding optimisation goal, where the red arrow indicates the different causal relationships between $X$ and $Y$. Left is CausalNL/InstanceGM [50, 11], middle is NPC [3] and right is ours.

**Clean label prior.** Our clean-label prior $p(Y)$ constrains the clean label to a set of label candidates for a particular training sample. Such label candidates change aims to 1) increase clean label coverage, and 2) represent uncertainty of the prior. Increase coverage improve the chances of including latent clean label in supervision. For noisy samples, increase the number of candidates in $p(Y)$ regularise noisy label training. Such dynamic prior distribution may resemble Mixup [53], label smoothing [28] or re-labeling [22] techniques that are commonly used in label noise learning. However, these approaches do not simultaneously follow the two design principles mentioned above. Mixup [53] and label smoothing [28] are effective approaches for designing soft labels for noisy label learning, but both aim to increase coverage, disregarding label uncertainty. Re-labeling switches the supervisory training signal to a more likely pseudo label, so it is very efficient, but it has limited coverage.

**Partial label learning** In partial label learning (PLL), each image is associated with a candidate label set defined as a partial label [36]. The goal of PLL is to predict the single true label associated with each training sample, assuming that the ground truth label is one of the labels in its candidate set. PICO [37] uses contrastive learning in an EM optimisation to address PLL. CAV [52] proposes class activation mapping to identify the true label within the candidate set. PRODEN [29] progressively identifies the true labels from a candidate set and updates the model parameter. The design of our informative clean label prior $p(Y)$ is inspired from PLL, but unlike PLL, there is no guarantee that the multiple label candidates in our prior contain the true label. Furthermore, the size of our candidate label set is determined by the probability that the training sample label is clean, where a low probability induces a prior with a large number of candidates for regularising training.

## 3 Method

We denote the noisy training set as $\mathcal{D} = \{(\mathbf{x}_i, \tilde{\mathbf{y}}_i)\}_{i=1}^{|\mathcal{D}|}$, where $\mathbf{x}_i \in \mathcal{X} \subset \mathbb{R}^{H \times W \times C}$ is the input image of size $H \times W$ with $C$ colour channels, $\tilde{\mathbf{y}}_i \in \mathcal{Y} \subset \{0, 1\}^{|\mathcal{Y}|}$ is the observed noisy label. We also have $\mathbf{y}$ as the unobserved clean label. We formulate our model with generative model that starts with the sampling of a label $\mathbf{y} \sim p(Y)$. This is followed by the clean-label conditioned generation of an image with $\mathbf{x} \sim p(X|Y = \mathbf{y})$, which are then used to produce the noisy label $\tilde{\mathbf{y}} \sim p(\tilde{Y}|Y = \mathbf{y}, X = \mathbf{x})$ (hereafter, we omit the variable names to simplify the notation). Below, in Sec. 3.1, we introduce our model and the optimisation goal. In Sec. 3.2 we describe how to construct informative prior, and the overall training algorithm is presented in Sec. 3.3.

### 3.1 Model

We aim to optimize the generative model $\log p(\mathbf{x}|\mathbf{y})$, which can be decomposed as follows:

$$\log p(\mathbf{x}|\mathbf{y}) = \log \frac{p(\tilde{\mathbf{y}}, \mathbf{y}, \mathbf{x})}{p(\tilde{\mathbf{y}}|\mathbf{x}, \mathbf{y})p(\mathbf{y})}. \tag{4}$$

In (4), $p(\mathbf{y})$ represents the prior distribution of the latent clean label. The optimisation of $p(\mathbf{x}|\mathbf{y})$ can be achieved by introducing a variational posterior distribution $q(\mathbf{y}|\mathbf{x})$, with:

$$\log p(\mathbf{x}|\mathbf{y}) = \log \frac{p(\tilde{\mathbf{y}}, \mathbf{y}, \mathbf{x})}{q(\mathbf{y}|\mathbf{x})} + \log \frac{q(\mathbf{y}|\mathbf{x})}{p(\tilde{\mathbf{y}}|\mathbf{x}, \mathbf{y})p(\mathbf{y})},$$
$$\mathbb{E}_{q(\mathbf{y}|\mathbf{x})}\left[\log p(\mathbf{x}|\mathbf{y})\right] = \mathbb{E}_{q(\mathbf{y}|\mathbf{x})}\left[\log \frac{p(\tilde{\mathbf{y}}, \mathbf{y}, \mathbf{x})}{q(\mathbf{y}|\mathbf{x})}\right] + \mathsf{KL}\left[q(\mathbf{y}|\mathbf{x})\|p(\tilde{\mathbf{y}}|\mathbf{x}, \mathbf{y})p(\mathbf{y})\right], \tag{5}$$

where $\mathsf{KL}[.]$ denotes the KL divergence, and

$$\mathbb{E}_{q(\mathbf{y}|\mathbf{x})}\left[\log \frac{p(\tilde{\mathbf{y}}, \mathbf{y}, \mathbf{x})}{q(\mathbf{y}|\mathbf{x})}\right] = \mathbb{E}_{q(\mathbf{y}|\mathbf{x})}\left[\log p(\tilde{\mathbf{y}}|\mathbf{x}, \mathbf{y})\right] + \mathbb{E}_{q(\mathbf{y}|\mathbf{x})}\left[\log \frac{p(\mathbf{x}|\mathbf{y})p(\mathbf{y})}{q(\mathbf{y}|\mathbf{x})}\right]. \tag{6}$$

Based on Eq. (5) and (6), the expected log likelihood of $p(\mathbf{x}|\mathbf{y})$ is defined as

$$\mathbb{E}_{q(\mathbf{y}|\mathbf{x})}\left[\log p(\mathbf{x}|\mathbf{y})\right] = \mathbb{E}_{q(\mathbf{y}|\mathbf{x})}\left[\log p(\tilde{\mathbf{y}}|\mathbf{x}, \mathbf{y})\right] - \mathsf{KL}\left[q(\mathbf{y}|\mathbf{x})\|p(\mathbf{x}|\mathbf{y})p(\mathbf{y})\right] + \mathsf{KL}\left[q(\mathbf{y}|\mathbf{x})\|p(\tilde{\mathbf{y}}|\mathbf{x}, \mathbf{y})p(\mathbf{y})\right]. \tag{7}$$

In Eq. (7), we parameterise $q(\mathbf{y}|\mathbf{x})$ and $p(\tilde{\mathbf{y}}|\mathbf{x}, \mathbf{y})$ with neural networks, as depicted in Figure 2. The generative model $p(\mathbf{x}|\mathbf{y})$ usually requires to model infinite number of samples based on conditional label and a generative model is hard to capture such relationship. However, since noisy label learning is a discriminative task and classification performance is our primary goal, the generation can be approximated with with finite training samples, which is given training set. More specifically, we defines $p(\mathbf{x}|\mathbf{y})$ only on data points $\{\mathbf{x}_i\}_{i=1}^{|\mathcal{D}|}$ by maximising $-\mathsf{KL}\left[q(\mathbf{y}|\mathbf{x})\|p(\mathbf{x}|\mathbf{y})p(\mathbf{y})\right]$ for a fixed $q(\mathbf{y}|\mathbf{x})$, with the optimum achieved by:

$$p(\mathbf{x}|\mathbf{y}) = \frac{q(\mathbf{y}|\mathbf{x})}{\sum_{i=1}^{|\mathcal{D}|} q(\mathbf{y}|\mathbf{x}_i)}. \tag{8}$$

Hence, the generative conditional $p(\mathbf{x}|\mathbf{y})$ can only represent the values of $\mathbf{x}$ within training set given the latent labels in $\mathbf{y}$. This allow us transform discriminative model into implicit generative model without additional computation cost.

### 3.2 Informative prior based on partial label learning

In Eq. (7), the clean label prior $p(\mathbf{y})$ is required. As mentioned in Sec. 2, we formulate $p(\mathbf{y})$ inspired from PLL [29, 37, 52]. However, it is worth noting that PLL has the partial label information available from the training set, while we have to dynamically build it during training. Therefore, the clean label prior $p(\mathbf{y})$ for each training sample is designed so that the hidden clean label has a high probability of being selected during most of the training. On one hand, we aim to have as many label candidates as possible during the training to increase the chances that $p(\mathbf{y})$ has a non-zero probability for the latent clean label. On the other hand, including all labels as candidates is a trivial solution that does

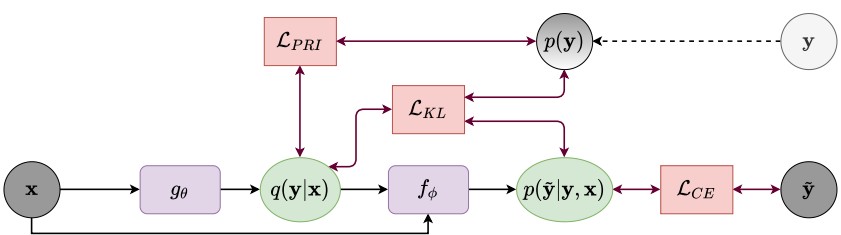

Figure 2: Training pipeline of our method. Shaded variables $\mathbf{x}$ and $\tilde{\mathbf{y}}$ are visible, and unshaded variable $\mathbf{y}$ is latent. $p(\mathbf{y})$ is constructed to approximate $\mathbf{y}$.

not represent a meaningful clean label prior. These two seemingly contradictory goals target the maximisation of label coverage and minimisation of label uncertainty, defined by:

$$\text{Coverage} = \frac{1}{|\mathcal{D}|} \sum_{i=1}^{|\mathcal{D}|} \sum_{j=1}^{|\mathcal{Y}|} \mathbb{1}\left(\mathbf{y}_i(j) \times p_i(j) > 0\right), \text{ and Uncertainty} = \frac{1}{|\mathcal{D}|} \sum_{i=1}^{|\mathcal{D}|} \sum_{j=1}^{|\mathcal{Y}|} \mathbb{1}\left(p_i(j) > 0\right),$$
(9)

where $\mathbb{1}(.)$ is the indicator function. In (9), coverage increases by approximating $p(Y)$ to a uniform distribution, but uncertainty is minimised when the clean label $\mathbf{y}_i$ is assigned maximum probability. In general, training samples for which the model is certain about the clean label, should have $p(\mathbf{y}_i) = 1$, while training samples for which the model is uncertain about the clean label, should have $p(\mathbf{y}_i) < 1$ with other candidate labels with probability $> 0$. Therefore, the clean label prior is defined by:

$$p_i(j) = \frac{\tilde{\mathbf{y}}_i(j) + \mathbf{c}_i(j) + \mathbf{u}_i(j)}{Z},$$
(10)

where $Z$ is a normalisation factor to make $\sum_{j=1}^{|\mathcal{Y}|} p_i(j) = 1$, $\tilde{\mathbf{y}}_i$ is the noisy label in the training set, $\mathbf{c}_i$ denotes the label to increase coverage, and $\mathbf{u}_i$ represents the label to increase uncertainty, both defined below. Motivated by the early learning phenomenon [25], where clean labels tend to be fit earlier in the training than the noisy labels, we maximise coverage by sampling from a moving average of model prediction for each training sample $\mathbf{x}_i$ at iteration $t$ with:

$$\mathcal{C}_i^{(t)} = \beta \times \mathcal{C}_i^{(t-1)} + (1 - \beta) \times \bar{\mathbf{y}}_i^{(t)},$$
(11)

where $\beta \in [0, 1]$ and $\bar{\mathbf{y}}^{(t)}$ is the softmax output from the model that predicts the clean label from the data input $\mathbf{x}_i$. For Eq. (11), $\mathcal{C}_i^{(t)}$ denotes the categorical distribution of the most likely labels for the $i^{th}$ training sample, which can be used to sample the one-hot label $\mathbf{c}_i \sim \mathsf{Cat}(\mathcal{C}_i^{(t)})$. The minimisation of uncertainty depends on our ability to detect clean-label and noisy-label samples. For clean samples, $p(\mathbf{y}_i)$ should converge to a one-hot distribution, maintaining the label prior focused on few candidate labels. For noisy samples, $p(\mathbf{y}_i)$ should be close to a uniform distribution to keep a large coverage of candidate labels. To compute the probability $w_i \in [0, 1]$ that a sample contains clean label, we use the sample selection approaches based on the unsupervised classification of loss values [22]. Then the label $\mathbf{u}_i$ is obtained by sampling from a uniform distribution of all possible labels proportionally to its probability of representing a noisy-label sample, with

$$\mathbf{u}_i \sim \mathcal{U}\left(\mathcal{Y}, \mathsf{round}(|\mathcal{Y}| \times (1 - w_i))\right),$$
(12)

where $\mathsf{round}(|\mathcal{Y}| \times (1 - w_i))$ represents the number of samples to be drawn from the uniform distribution rounded up to the closest integer.

### 3.3 Training

We can now return to the optimisation of Eq. (7), where we define the neural networks $g_\theta : \mathcal{X} \to \Delta^{|\mathcal{Y}|-1}$ that outputs the categorical distribution for the clean label in the probability simplex space $\Delta^{|\mathcal{Y}|-1}$ given an image $\mathbf{x} \in \mathcal{X}$, and $f_\phi : \mathcal{X} \times \Delta^{|\mathcal{Y}|-1} \to \Delta^{|\mathcal{Y}|-1}$ that outputs the categorical distribution for the noisy training label given an image and the clean label distribution from $g_\theta(.)$. The first term in the right-hand side (RHS) in Eq. (7) is optimised with the cross-entropy loss:

$$\mathcal{L}_{CE}(\theta, \phi, \mathcal{D}) = \frac{1}{|\mathcal{D}| \times K} \sum_{(\mathbf{x}_i, \tilde{\mathbf{y}}_i) \in \mathcal{D}} \sum_{j=1}^{K} \ell_{CE}(\tilde{\mathbf{y}}_i, f_\phi(\mathbf{x}_i, \hat{\mathbf{y}}_{i,j})).$$
(13)

where $\{\hat{\mathbf{y}}_{i,j}\}_{j=1}^K \sim \mathsf{Cat}(g_\theta(\mathbf{x}_i))$, with $\mathsf{Cat}(.)$ denoting a categorical distribution. The second term in the RHS in Eq. (7) uses the estimation of $p(\mathbf{x}|\mathbf{y})$ from Eq. (8) to optimise the KL divergence:

$$\mathcal{L}_{PRI}(\theta,\mathcal{D}) = \frac{1}{|\mathcal{D}|} \sum_{(\mathbf{x}_i,\tilde{\mathbf{y}}_i)\in\mathcal{D}} \mathsf{KL}\left[g_\theta(\mathbf{x}_i)\Big\|c_i \times \frac{g_\theta(\mathbf{x}_i)}{\sum_j g_\theta(\mathbf{x}_j)} \odot \mathbf{p}_i\right], \tag{14}$$

where $\mathbf{p}_i = [p_i(j=1),...,p_i(j=|\mathcal{Y}|)] \in \Delta^{|\mathcal{Y}|-1}$ is the clean label prior defined in Eq. (10), $c_i$ is a normalisation factor, and $\odot$ is the element-wise multiplication. The last term in the RHS of Eq. (7) is the KL divergence between $q(\mathbf{y}|\mathbf{x})$ and $p(\tilde{\mathbf{y}}|\mathbf{x},\mathbf{y})p(\mathbf{y})$, which represents the gap between $\mathbb{E}_{q(\mathbf{y}|\mathbf{x})}\left[\log p(\mathbf{x}|\mathbf{y})\right]$ and $\mathbb{E}_{q(\mathbf{y}|\mathbf{x})}\left[\log\frac{p(\tilde{\mathbf{y}},\mathbf{y},\mathbf{x})}{q(\mathbf{y}|\mathbf{x})}\right]$. According to the expectation-maximisation (EM) derivation [8, 18], the smaller this gap, the better $q(\mathbf{y}|\mathbf{x})$ approximates the true posterior $p(\mathbf{y}|\mathbf{x})$, so the loss function associated with this third term is:

$$\mathcal{L}_{KL}(\theta,\phi,\mathcal{D}) = \frac{1}{|\mathcal{D}|} \sum_{(\mathbf{x}_i,\tilde{\mathbf{y}}_i)\in\mathcal{D}} \mathsf{KL}\left[g_\theta(\mathbf{x}_i)\Big\|f_\phi(\mathbf{x}_i, g_\theta(\mathbf{x}_i)) \odot \mathbf{p}_i\right]. \tag{15}$$

Our final loss to minimise is

$$\mathcal{L}(\theta,\phi,\mathcal{D}) = \mathcal{L}_{CE}(\theta,\phi,\mathcal{D}) + \mathcal{L}_{PRI}(\theta,\mathcal{D}) + \mathcal{L}_{KL}(\theta,\phi,\mathcal{D}). \tag{16}$$

After training, a test image $\mathbf{x}$ is associated with a class with $g_\theta(\mathbf{x})$. An interesting point about this derivation is that the implicit approximation of $p(\mathbf{x}|\mathbf{y})$ enables the minimisation of the loss in (16) using regular stochastic gradient descent instead of a more computationally complex $EM$ algorithm [33].

## 4    Experiments

We show experimental results on instance-dependent synthetic and real-world label noise benchmarks with datasets CIFAR10/100 [20]. We also test on three instance-dependent real-world label noise datasets, namely: Animal-10N [35], Red Mini-ImageNet [15], and Clothing1M [46].

### 4.1    Datasets

**CIFAR10/100** [20] contain a training set with 50K images and a testing of 10K images of size 32 $\times$ 32 $\times$ 3, where CIFAR10 has 10 classes and CIFAR100 has 100 classes. We follow previous works [44] and synthetically generate instance-dependent noise (IDN) with rates in {0.2, 0.3, 0.4 ,0.5}. **CIFAR10N/CIFAR100N** is proposed by [43] to study real-world annotations for the original CIFAR10/100 images and we test our framework on {aggre, random1, random2, random3, worse} types of noise on CIFAR10N and {noisy} on CIFAR100N. **Red Mini-ImageNet** is a real-world dataset [15] containing 100 classes, each containing 600 images from ImageNet, where images are resized to 32 $\times$ 32 pixels from the original 84 $\times$ 84 to enable a fair comparison with other baselines [48]. **Animal 10N** [35] is a real-world dataset containing 10 animal species with five pairs of similar appearances (wolf and coyote, etc.). The training set size is 50K and testing size is 10K, where we follow the same set up as [5]. **Clothing1M** is a real-world dataset with 100K images and 14 classes. The labels are automatically generated from surrounding text with an estimated noise ratio of 38.5%. The dataset also contains clean samples for training and validation but we only use clean test for measuring model performance.

### 4.2    Practical considerations

We follow commonly used experiment setups for all benchmarks described in Sec. 4.1. [1] For the hyper-parameter setup, $K$ in (13) is set to 1, and $\beta$ in Eq. (11) is set to 0.9. For $w$ in Eq. (12), we follow the commonly used Gaussian Mixture Model (GMM) unsupervised classification from [22]. For warmup epochs, $w$ is randomly generated from a uniform distribution. Note that the approximation of the generative model from (8) is done within each batch, not the entire the dataset. Also, the minimisation of $\mathcal{L}_{PRI}(.)$ can be done with the reversed KL using $\mathsf{KL}\left[c_i \times \frac{g_\theta(\mathbf{x}_i)}{\sum_j g_\theta(\mathbf{x}_j)} \odot \mathbf{p}_i \Big\| g_\theta(\mathbf{x}_i)\right]$.

---

[1] Please see the supplementary material about implementation details.

| Method | CIFAR10 | | | |
|---|---|---|---|---|
| | 20% | 30% | 40% | 50% |
| CE | 86.93±0.17 | 82.42±0.44 | 76.68±0.23 | 58.93± 1.54 |
| DMI [47] | 89.99± 0.15 | 86.87± 0.34 | 80.74± 0.44 | 63.92±3.92 |
| Forward [32] | 89.62±0.14 | 86.93±0.15 | 80.29±0.27 | 65.91±1.22 |
| CoTeaching [12] | 88.43±0.08 | 86.40±0.41 | 80.85±0.97 | 62.63± 1.51 |
| TMDNN [49] | 88.14± 0.66 | 84.55±0.48 | 79.71±0.95 | 63.33± 2.75 |
| PartT [44] | 89.33± 0.70 | 85.33±1.86 | 80.59±0.41 | 64.58± 2.86 |
| kMEIDTM [6] | 92.26± 0.25 | 90.73± 0.34 | 85.94± 0.92 | 73.77±0.82 |
| CausalNL [50] | 81.47± 0.32 | 80.38± 0.44 | 77.53± 0.45 | 67.39±1.24 |
| Ours | **92.65±0.13** | **91.96±0.20** | **91.02±0.44** | **89.94±0.45** |

Table 1: Accuracy (%) on the test set for CIFAR10-IDN. Most results are from [6]. Experiments are repeated 3 times to compute mean±standard deviation. Top part shows discriminative and bottom shows generative models. Best results are highlighted.

| Method | CIFAR100 | | | |
|---|---|---|---|---|
| | 20% | 30% | 40% | 50% |
| CE | 63.94±0.51 | 61.97±1.16 | 58.70±0.56 | 56.63±0.69 |
| DMI [47] | 64.72±0.64 | 62.8±1.46 | 60.24±0.63 | 56.52±1.18 |
| Forward [32] | 67.23±0.29 | 65.42±0.63 | 62.18±0.26 | 58.61±0.44 |
| CoTeaching [12] | 67.40±0.44 | 64.13±0.43 | 59.98±0.28 | 57.48±0.74 |
| TMDNN [49] | 66.62±0.85 | 64.72±0.64 | 59.38±0.65 | 55.68±1.43 |
| PartT [44] | 65.33±0.59 | 64.56±1.55 | 59.73±0.76 | 56.80±1.32 |
| kMEIDTM [6] | 69.16±0.16 | 66.76±0.30 | 63.46±0.48 | 59.18±0.16 |
| CausalNL [50] | 41.47±0.43 | 40.98±0.62 | 34.02±0.95 | 32.13±2.23 |
| Ours | **71.24±0.43** | **69.64±0.78** | **67.48±0.85** | **63.60±0.17** |

Table 2: Accuracy (%) on the test set for CIFAR100-IDN. Most results are from [6]. Experiments are repeated 3 times to compute mean±standard deviation. Top part shows discriminative and bottom shows generative models. Best results are highlighted.

| Method | CIFAR10N | | | | | CIFAR100N |
|---|---|---|---|---|---|---|
| | Aggregate | Random 1 | Random 2 | Random 3 | Worst | Noisy |
| CE | 87.77±0.38 | 85.02±0.65 | 86.46±1.79 | 85.16±0.61 | 77.69±1.55 | 55.50±0.66 |
| Forward T [32] | 88.24±0.22 | 86.88±0.50 | 86.14±0.24 | 87.04±0.35 | 79.79±0.46 | 57.01±1.03 |
| T-Revision [45] | 88.52±0.17 | 88.33±0.32 | 87.71±1.02 | 80.48±1.20 | 80.48±1.20 | 51.55±0.31 |
| Positive-LS [28] | 91.57±0.07 | 89.80±0.28 | 89.35±0.33 | 89.82±0.14 | 82.76±0.53 | 55.84±0.48 |
| F-Div [42] | 91.64±0.34 | 89.70±0.40 | 89.79±0.12 | 89.55±0.49 | 82.53±0.52 | 57.10±0.65 |
| Negative-LS [41] | 91.97±0.46 | 90.29±0.32 | 90.37±0.12 | 90.13±0.19 | 82.99±0.36 | 58.59±0.98 |
| CORES$^2$ [7] | 91.23±0.11 | 89.66±0.32 | 89.91±0.45 | 89.79±0.50 | 83.60±0.53 | **61.15±0.73** |
| VolMinNet [23] | 89.70±0.21 | 88.30±0.12 | 88.27±0.09 | 88.19±0.41 | 80.53±0.20 | 57.80±0.31 |
| CAL [55] | 91.97±0.32 | 90.93±0.31 | 90.75±0.30 | 90.74±0.24 | 85.36±0.16 | **61.73±0.42** |
| Ours | **92.57±0.20** | **91.97±0.09** | **91.42±0.06** | **91.83±0.12** | **86.99±0.36** | 61.54±0.22 |

Table 3: Accuracy (%) on the test set for CIFAR10N/100N. Results are taken from [43] using methods containing a single classifier with ResNet-34. Best results are highlighted.

This reversed KL divergence also provides solutions where the model and implied posterior are close. In fact, the KL and reversed KL losses are equivalent when $\sum_j g_\theta(\mathbf{x}_j)$ has a uniform distribution over the classes in $\mathcal{Y}$ and the prior $\mathbf{p}_i$ is uniform in the negative labels. We tried the optimisation using both versions of the KL divergence (i.e., the one in (14) and the one above in this section), with the reversed one generally producing better results, as shown in the ablation study in Sec. 4.4. For all experiments in Sec. 4.3, we rely on the reversed KL loss. For the real-world datasets Animal-10N, Red Mini-ImageNet and Clothing1M we also test our model with the training and testing of an ensemble of two networks. Our code is implemented in Pytorch and experiments are performed on RTX 3090.

## 4.3 Experimental Results

**Synthetic benchmarks.** The experimental results of our method with IDN problems on CIFAR10/100 are shown in Tab.1 and Tab.2. Compared with the previous SOTA kMEDITM [6], on CIFAR10, we

| Method | Noise rate | | | | Method | Accuracy |
|---|---|---|---|---|---|---|
| | 0.2 | 0.4 | 0.6 | 0.8 | CE | 79.4 |
| CE | 47.36 | 42.70 | 37.30 | 29.76 | SELFIE [35] | 81.8 |
| Mixup [53] | 49.10 | 46.40 | 40.58 | 33.58 | JoCoR [40] | 82.8 |
| DivideMix [22] | 50.96 | 46.72 | 43.14 | 34.50 | PLC [54] | 83.4 |
| MentorMix [14] | 51.02 | 47.14 | 43.80 | 33.46 | Nested + Co-T [5] | 84.1 |
| FaMUS [48] | 51.42 | 48.06 | 45.10 | 35.50 | InstanceGM [11] | 84.6 |
| Ours | 53.34 | 49.56 | 44.08 | 36.70 | Ours | 82.7 |
| Ours ensemble | **57.56** | **52.68** | **47.12** | **39.54** | Ours ensemble | **85.7** |

Table 4: Test accuracy (%) on Red Mini-ImageNet (Left) with different noise rates and baselines from FaMUS [48], and on Animal-10N (Right), with baselines from [5]. Best results are highlighted.

| CE | Forward [32] | PTD-R-V [44] | ELR [26] | kMEIDTM [6] | CausalNL [50] | Our ensemble |
|---|---|---|---|---|---|---|
| 68.94 | 69.84 | 71.67 | 72.87 | 73.34 | 72.24 | **74.35** |

Table 5: Test accuracy (%) on the test set of Clothing1M. Results are obtained from their respective papers. We only use the noisy training set for training. Best results are highlighted.

achieve competitive performance on low noise rates and up to 16% improvements for high noise rates. For CIFAR100, we consistently improve 2% to 4% in all noise rates. Compared with the previous SOTA generative model CausalNL [50], our improvement is significant for all noise rates. The superior performance of our method indicates that our implicit generative modelling and clean label prior construction is effective when learning with label noise.

**Real-world benchmarks.** In Tab.3, we show the performance of our method on the CIFAR10N/100N benchmark. Compared with other single-model baselines, our method achieves at least 1% improvement on all noise rates on CIFAR10N, and it has a competitive performance on CIFAR100N. The Red Mini-ImageNet results in Tab.4 (left) show that our method achieves SOTA results for all noise rates with 2% improvements using a single model and 6% improvements using the ensemble of two models. The improvement is substantial compared with previous SOTA FaMUS [48] and DivideMix [22]. In Tab.4(right), our single-model result on Animal-10N achieves 1% improvement with respect to the single-model SELFIE [35]. Considering our approach with an ensemble of two models, we achieve a 1% improvement over the SOTA Nested+Co-teaching [5]. Our ensemble-model result on Clothing1M in Tab.5 shows a competitive performance of 74.4%, which is 2% better than the previous SOTA generative model CausalNL [50].

## 4.4 Analysis

**Ablation** The ablation analysis of our method is shown in Tab.6 with the IDN problems on CIFAR10. First row ($\mathcal{L}_{CE}$) shows the results of the training with a cross-entropy loss using the training samples and labels in $\mathcal{D}$. The second row ($\mathcal{L}_{CE} + \mathcal{L}_{CE\_PRI} + \mathcal{L}_{KL}$) shows the result of our method, replacing the KL divergence in $\mathcal{L}_{PRI}$ as defined in (14), by a soft version of cross entropy loss. Next, the third row ($\mathcal{L}_{CE} + \mathcal{L}_{PRI} + \mathcal{L}_{KL}$) shows our method with the loss defined in (16). As mentioned in Sec. 4.2, these two forms provides similar solution where the model and implicit posterior are close and $\mathcal{L}_{PRI}$ reverse generally performs better. In the fourth row ($\mathcal{L}_{CE} + \mathcal{L}_{PRI}$ reversed) by optimising the lower bound to $\mathbb{E}_{q(\mathbf{y}|\mathbf{x})}[\log p(\mathbf{x}|\mathbf{y})]$ and finally the last row by optimising the whole objective function from (16) in the last row ($\mathcal{L}_{CE} + \mathcal{L}_{PRI}$ reversed $+ \mathcal{L}_{KL}$ (Ours)). In general, notice that the reversed $\mathcal{L}_{PRI}$ improves the results; the KL divergence in $\mathcal{L}_{PRI}$ works better than the CE loss; and the optimisation of the whole loss in (16) is better than optimising the lower bound, which justifies the inclusion of $\mathcal{L}_{KL}(.)$ in the loss.

**Coverage and uncertainty visualisation** We visualise coverage and uncertainty from Eq. (9) at each training epoch for IDN CIFAR10/100 and CIFAR10N setups. In all cases, label coverage increases as training progresses, indicating that our prior tends to always cover the clean label. In fact, coverage reaches nearly 100% for CIFAR10 at 20% IDN and 97% for 50% IDN. Furthermore, for CIFAR100 at 50% IDN, we achieve 82% coverage, and for CIFAR10N "worse", we reach 92% coverage. In terms of uncertainty, we notice a steady reduction as training progresses for all problems, where the uncertainty values tend to be slightly higher for the problems with higher noise rates and more classes. For instance, uncertainty is between 2 and 3 for the for CIFAR10's IDN benchmarks, increasing to be

| Method | CIFAR10 | | | |
|---|---|---|---|---|
| | 20% | 30% | 40% | 50% |
| $\mathcal{L}_{CE}$ | 86.93 | 82.42 | 76.68 | 58.93 |
| $\mathcal{L}_{CE} + \mathcal{L}_{CE\_PRI} + \mathcal{L}_{KL}$ | 85.96 | 82.74 | 78.34 | 73.72 |
| $\mathcal{L}_{CE} + \mathcal{L}_{PRI} + \mathcal{L}_{KL}$ | 91.36 | 90.88 | 90.25 | 88.77 |
| $\mathcal{L}_{CE} + \mathcal{L}_{PRI}$ reversed | 92.40 | 90.23 | 87.75 | 80.46 |
| $\mathcal{L}_{CE} + \mathcal{L}_{PRI}$ reversed + $\mathcal{L}_{KL}$ (Ours) | 92.65 | 91.96 | 91.02 | 89.94 |

Table 6: Ablation analysis of our proposed method. Please see text for details.

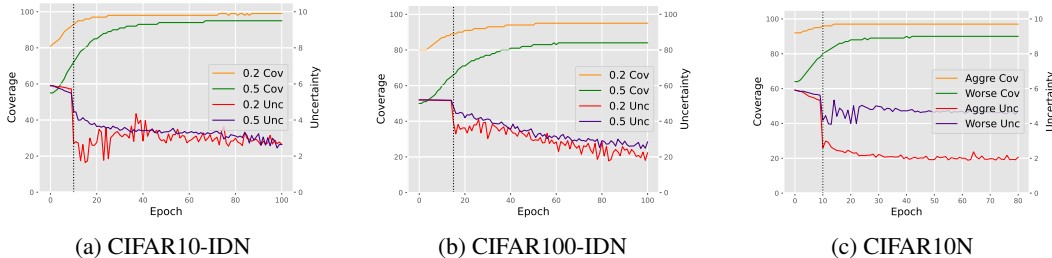

| (a) CIFAR10-IDN | (b) CIFAR100-IDN | (c) CIFAR10N |
|---|---|---|

Figure 3: Coverage (Cov) and uncertainty (Unc) for (a) CIFAR10-IDN (20% and 50%), (b) CIFAR100-IDN (20% and 50%), and (c) CIFAR10N ("Worse" and "Aggre"). Y-axis shows coverage (left) and uncertainty (right). The dotted vertical line indicates the end of warmup training.

| | CE | DivideMix [22] | CausalNL [50] | InstanceGM [11] | Ours |
|---|---|---|---|---|---|
| CIFAR | 2.1h | 7.1h | 3.3h | 30.5h | 2.3h |
| Clothing1M | 4h | 14h | 10h | 43h | 4.5h |

Table 7: Running times of various methods on CIFAR100 with 50% IDN and Clothing1M using the hardware listed in Sec. 4.2.

between 2 and 4 for CIFAR10N. For CIFAR100's IDN benchmarks, uncertainty is between 20 and 30. These results suggest that our prior clean label distribution is effective at selecting the correct clean label while reducing the number of label candidates during training.

**Training time comparison** One of the advantages of our approach is its efficient training algorithm, particularly when compared with other generative and discriminative methods. Tab. 7 shows the training time for competing approaches on CIFAR100 with 50% IDN and Clothing1M using the hardware specified in Sec. 4.2 . In general, our method has a smaller training time than competing approaches, being $1.4\times$ faster than CausalNL [50], $3\times$ faster than DivideMix [22], and and $13\times$ faster than InstanceGM [11].

## 5 Conclusion

In this paper, we presented a new learning algorithm to optimise a generative model represented by $p(X|Y)$ that directly associates data and clean labels instead of maximising the joint data likelihood, denoted by $p(X, \tilde{Y})$. Our optimisation implicitly estimates $p(X|Y)$ with the discriminative model $q(Y|X)$ eliminating the inefficient generative model training. Furthermore, we introduce an informative label prior for maintaining high coverage of latent clean label and regularise noisy label training. Results on synthetic and real-world noisy-label benchmarks show that our generative method has SOTA results, but with complexity comparable to discriminative models.

A limitation of the proposed method that needs further exploration is a comprehensive study of the model for $q(Y|X)$. In fact, the competitive results shown in this paper are obtained from fairly standard models for $q(Y|X)$ without exploring sophisticated noisy-label learning techniques. In the future, we will use more powerful models for $q(Y|X)$. Another issue of our model is the difficulty to estimate $p(X|Y)$ in real-world datasets containing images of high resolution. We will study more adequate ways to approximate $p(X|Y)$ in such scenario using data augmentation strategies to increase the scale of the dataset.

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
