# OpenReview forum: "Generative Noisy-Label Learning by Implicit Dicriminative Approximation with Partial Label Prior"
_NeurIPS.cc/2023/Conference — Submitted to NeurIPS 2023_

### Official Review · Reviewer_jVwM · 2023-07-02

**Soundness:** 3 good
**Presentation:** 2 fair
**Contribution:** 2 fair
**Rating:** 6
**Confidence:** 4

**Summary:**

This paper introduces a novel approach to tackle the challenge of noisy label learning through a generative framework. Firstly, it presents a new model optimization technique that establishes a direct association between the data and clean labels. Secondly, the generative model is implicitly estimated by leveraging a discriminative model, thereby eliminating the need for training a separate generative model and enhancing efficiency. Thirdly, the paper proposes an informative label prior inspired by partial label learning, which serves as a supervision signal for noisy label learning. Extensive experiments conducted on various noisy-label benchmarks demonstrate that the proposed generative model achieves state-of-the-art results. Remarkably, it achieves these results while maintaining a comparable computational complexity to discriminative models.

**Strengths:**

- This paper introduces an informative prior for the latent clean label in noisy label learning, which is interesting.
- Experimental results show the effectiveness of the proposed method.


**Weaknesses:**

1. My concern is whether it is reasonable to build a generative noisy-label learning model, which only assumes that Y causes X. The previous work, where the latent feature Z and Y cause X, seems more reasonable.
2. From the perspective of the algorithm, it seems unnecessary to limit the algorithm to image data, while the notation at Line 138, Page 4 and the experiments focus on the image data, which is required to be explained or conduct more experiments on other kinds of datasets.
3. Some symbols need to be explained. For example, at Eq.(9), Page 5, $y_i(j)$ and $p_i(j)$ is very confusing. So is $|\mathcal{Y}|$ at Eq.(12), Page 5.
4. Some paragraphs, especially for the approach, need to be polished up to better explain how to address the proposed three issues in the abstract.


**Questions:**

What is the practical scene of such a generative noisy-label learning model?

**Limitations:**

The authors analyze the limitations in the conclusion of the paper. It is suggested that more explanations should be made on the reasonability of their generative model. Besides, some paragraphs need to be polished up.

---

> ### Author Rebuttal · Authors · 2023-08-03
>
> We thank Reviewer jVwM for the insightful comments.
>
> > Z and Y cause X seems more reasonable
>
> This is a reasonable concern. In principle, the latent feature $Z$ is important for generating $X$ in a causal relationship because the generative model would need $Z$ to "anchor" the $P(X|Y)$ modeling as there are infinite number of images that can be generated from the label. However, for the classification task, the generation ability of image does not help improve classification performance and  latent feature $Z$ is not necessary. Furthermore, modeling $Z$ would require an extra module and extra computational resources.
>
> Hence, we do not need $Z$ to anchor the image generation and instead represent $p(X|Y)$ with the discriminative model $q(Y|X)$, as explained in Eq.8.  This means that we do not have an image generator in our model.
>
> > Different kinds of datasets.
>
> That is an excellent point, thanks for that. We have tested our method on two public NLP news topic classification tasks. Please see the attached PDF Table 2.
>
> We selected DyGEN [1] as a recently proposed generative noisy label method as baseline. We followed the same hyper-parameter setup and architecture, and compared with all baselines. Our method outperforms other approaches in the NLP tasks.
>
> > Symbols to be explained.
>
> - $\mathbf{y_i}(j)$ is the $j_{th}$ label of one-hot latent clean label $\mathbf{y_i}$.
> - $\mathbf{p_i}(j)$ is the $j_{th}$ label of one-hot clean label prior $\mathbf{p_i}$.
> - Eq. 9: "Coverage" defines if latent clean label $\mathbf{y_i}$ is present in $\mathbf{p_i}$. "Uncertainty" defines how many labels are present in $\mathbf{p_i}$.
> - Eq. 12: $|\mathcal{Y}|$ is the label space cardinality (i.e., number of labels).
>
> > Some paragraphs, especially for the approach, need to be polished up to better explain how to address the proposed three issues in the abstract.
>
> We will polish the approach with our arguments used in rebuttal. Here we listed the questions we proposed in abstract: Q1: Previous generative model optimizes the joint likelihood. Q2: Generative models are challenging to train. Q3: Uninformative clean label prior.
>
> For question Q1 and Q2, this is due to intractability of direct optimization $P(X|Y)$ because the infinite number of samples can be generated from labels. In Eq.8, we define $P(X|Y)$ only on the finite number of training samples given by classification task. This makes directly optimise $P(X|Y)$ possible and solve Q1. Furthermore, this allows us to optimise standard discriminative model in generative goal, which solves Q2.
>
>  For Q3, the variational posterior $q(Y|X)$ depends on the modeling of latent clean label $Y$. Motivated by [3], we represent $Y$ with a partial label distribution $P(Y)$ and define clean-label coverage and uncertainty for constructing informativeness partial label, which solves Q3.
>
> > Practical scene of generative noisy-label learning
>
> - Generative noisy label learning is a promising way of modeling the label transition matrix, as discussed by CausalNL[2].
> - Our proposed method improves upon this idea by simplifying the generative part of the method and significantly improve classification accuracy.
> - Our method combines generative modeling, noisy label learning and partial label learning in a unified framework.
>
> ***
> [1] DyGen: Learning from Noisy Labels via Dynamics-Enhanced Generative Modeling, KDD 2023
>
> [2] Instance-dependent Label-noise Learning under a Structural Causal Model, NeurIPS 2021
>
> [3] Decompositional Generation Process for Instance-Dependent Partial Label Learning, ICLR 2023

---

> > ### Comment · Reviewer_jVwM · 2023-08-13
> > **About the rebuttal**
> >
> > 1. The authors have address most of my concerns.
> >
> > 2. Could you provide some evidence about the argument in the rebuttal that "for the classification task, the generation ability of image does not help improve classification performance and latent feature is not necessary"？

---

> > > ### Author Response · Authors · 2023-08-13
> > > **Evidence of the argument**
> > >
> > > We thanks for Reviewer jVwM reply.
> > >
> > > > Evidence of image generation does not help improve classification performance.
> > >
> > > we refer to "semi-supervised with GAN" as a close research field that combines generative model and discriminative task.
> > > - [1] observed that feature matching generator obtains better semi-supervised performance while generate poor images.
> > > - [2] also observed that model generated better images but failed to improve semi-supervised performance.
> > > - [3] showed that a perfect generator (generating images that exactly matches the input distribution) does not improve generalization performance under semi-supervised setup.
> > >
> > > Although semi-supervised task is different from noisy label learning, they are both classification task that aims to find decision boundary. We believe this could serve as an evidence for "image generation does not help improve classification performance".
> > >
> > > > Evidence of latent feature is not necessary.
> > >
> > > There are other methods does not include $Z$ in their generative process (NPC[4], DyGEN [5]). As stated in [4]:
> > > - Modeling $Z$ for $X$ with large resolution leads to sub-optimal reconstruction, which is a common issue for generative modeling (VAE).
> > > - Because $Z$ and $Y$ jointly generate $X$, they need to be disentangled for classification task. And such disentanglement is not the main goal of noisy label classification.
> > >
> > > ---
> > > [1] Improved techniques for training gans, NeurIPS 2016
> > >
> > > [2] Adversarial generator-encoder networks, AAAI 2018
> > >
> > > [3] Good Semi-supervised Learning That Requires a Bad GAN, NeurIPS 2017
> > >
> > > [4] From Noisy Prediction to True Label: Noisy Prediction Calibration via Generative Model, ICML 2022
> > >
> > > [5] DyGen: Learning from Noisy Labels via Dynamics-Enhanced Generative Modeling, KDD 2023

---

> > > > ### Comment · Reviewer_jVwM · 2023-08-14
> > > >
> > > > Thanks for the response. I will consider raising the rating.

---

> > > > > ### Author Response · Authors · 2023-08-14
> > > > >
> > > > > Thanks for your valuable comments. We will include these reference in our updated version to strength our arguments.

---

### Official Review · Reviewer_rUmF · 2023-07-05

**Soundness:** 1 poor
**Presentation:** 2 fair
**Contribution:** 1 poor
**Rating:** 1
**Confidence:** 4

**Summary:**

This paper discusses the solution of learning with noisy labels by directly optimizing P(X|Y) relying on associating the data with clean labels directly. A informative label prior is derived with experimental results on several benchmark datasets.

**Strengths:**

The author derive a solution for generative noisy label learning under some very strong and unrealistic assumptions.

**Weaknesses:**

(1)	The paper is very difficult to understand partially because the method is not well motivated. For example, there are many generative model based noisy label learning methods such as
(i)	Label-Noise Robust Generative Adversarial Networks, CVPR 2019
(ii)	From Noisy Prediction to True Label: Noisy Prediction Calibration via Generative Model, ICML 2022.
Moreover, the approaches that donot directly optimize the P(X|Y) usually because it is not tractable under general conditions. This paper is adding many strong but unrealistic assumptions to directly optimize P(X|Y) may not sound like a reasonable approach. Namely the contribution point 1 is not clear.
(2)	I have strong doubt about the theoretical soundness of the paper. For instance, in the equation (10), why the clean label prior can be defined as the linear combination of the three terms \tilde{y}_i, c(i) and u_{i}(j), as these three terms may not be mutually exclusive and they may overlap with each other.

(3)	Another issue is in the equation (12), the authors said “ the label ui is obtained by sampling from a uniform distribution of all possible labels proportionally to its probability of representing a noisy-label sample”, this assumption is way too strong and cannot be true. As a matter of fact, there are very few cases that the label distributions are uniform. Frequently, the noisy label distributions are highly imbalanced. Please take a look at the reference [ii] in ICML 2022, the usual way to assume noisy label distribution is multinomial instead of uniform distribution.

(4)	The contribution is not clear. Besides the contribution 1 is not really a contribution, in the second point of their claimed contribution, they said “Our generative model is implicitly estimated with a discriminative model, making it computationally more efficient than previous generative approaches.”
This is also not valid as there are many generative adversarial network based noisy labeling which applied both generative model and discriminative model and let them collaborate with each other (for instance, the reference [i]) Thus, the second point is also very ordinary and not novel as well.



**Questions:**

There are many issues in this paper including poor theoretical ground, unrealistic and wrong assumption as well as very limited novelty.

**Limitations:**

As mentioned before, as the contribution of this paper is not clear. Both of the points (1) and (2) are actually done by previous work with a wider scope and less strict assumption.

In addition to that, assuming the noisy label distribution to be uniform is too restrictive, making the solution has little usage.

Last but not least, the solution is derived based on very strong and unrealistic assumption such as equations (10) and (12), making the experimental results not convincing.

In summary, the paper is too far away from the level of a Neurips paper.

---

> ### Author Rebuttal · Authors · 2023-08-02
>
> We disagree with most points from Reviewer rUmF and provide details below to support our position.
>
> > Strong but unrealistic assumptions to directly optimize $P(X|Y)$
>
> To explain the assumptions to optimise $P(X|Y)$, we first need to consider its intractability, which is in part due to the infinite number of image samples $\mathbf{x}$ that can be generated from their clean labels $\mathbf{y}$. One solution to mitigate such intractability is the use of a latent image representation $Z$ to "anchor" the image generation process ([CausalNL1] and InstanceGM[2]). However, note that such image generation process is in fact irrelevant for the noisy label classification task that we aim to solve. Image generation ability from noisy labels may be important for some papers, such as “Label-Noise Robust Generative Adversarial Networks, CVPR 2019” cited by Reviewer rUmF, but the goal of that paper is misaligned with our goal (noisy-label classification). Their experiments also show little relevance to our task.
>
> For the classification task that we address in our paper, we assume that $Z$ is unnecessary and that $P(X|Y)$ is defined only on the finite number of training samples given by  classification task. These assumptions facilitate the direct optimisation of $P(X|Y)$ and alleviate the problematic training of an image generator, particularly when $X$ is large (as noticed by Reviewer 9Aeh). All of our experiments, tested under synthetic and real-world datasets, show significantly improvement over previous generative and discriminative methods, both in terms of accuracy and efficiency.
>
> Given our arguments above and the superior experimental results, we respectfully disagree with Reviewer rUmF's comment that our assumption is "strong and unrealistic".
>
> > Why the clean label prior can be defined as the linear combination of the three terms.
>
> Given that clean labels are latent, our goal is to formulate an instance-wise partial label prior with multiple candidate labels that have high likelihood of containing the clean label. Motivated by [3], partial label can be factorized into one ground truth label and multiple complementary labels. For noisy label learning, we propose a formulation based on maximising label coverage and minimising label uncertainty, which is achieved with the combination of the training noisy label $\tilde{\mathbf{y_i}}$, the label coverage term $\mathbf{c_i}$, and an uncertainty term $\mathbf{u_i}$.
>
> > Three terms may overlap.
>
> The reviewer is correct in saying that these three terms may not be mutually exclusive and may overlap.  Because the overlapping likely means that the method found the clean label for the instance. In that case, our label coverage is maximised to full coverage and uncertainty is minimised to a one-hot label, which is the ideal case for clean label. It is unclear why this overlap would be a problem for the theoretical soundness of the paper.
>
> > Uniform noisy label distribution is wrong. Please take a look at the reference NPC[4] in ICML 2022, the usual way to assume noisy label distribution is multinomial instead of uniform distribution.
>
> We agree with Reviewer rUmF on this point. In fact, the label coverage term $\mathbf{c_i}$ in $p(\mathbf{y})$ is obtained by sampling from a multinomial distribution. This is the same as NPC[4] for noisy label distribution. The label uncertainty term $\mathbf{u_i}$ is sampled from a uniform label distribution to smooth the partial label prior based on the likelihood $w_i$ of the sample being noisy-labelled, where low $w_i$ implies higher uncertainty, producing a more uniform $p(\mathbf{y})$to better regularise the training process.
>
> > The claim that the generative model is implicitly estimated with a discriminative model is ordinary and not novel.
>
> As far as we are aware, **all** single-stage (i.e., end-to-end training) generative noisy-label learning methods previously proposed in the field depend on the generation of images or generation of low-dimensional image representations. Ours is the first single-stage generative noisy-label learning method that **only** depends on a discriminative model and a transition matrix, making the run-time complexity of our method similar to the complexity of discriminative models and much smaller than the complexity of previously proposed generative models. Please note in Table 7 that our method is as efficient as training a simple CE-loss approach, and extremely more efficient than SOTA discriminative models DivideMix[5] and generative models [1,2].
>
> ***
>
> [1]  Instance-dependent Label-noise Learning under a Structural Causal Model, NeurIPS 2021
>
> [2] Instance-dependent noisy label learning via graphical modelling, WACV 2022
>
> [3] Decompositional Generation Process for Instance-Dependent Partial Label Learning, ICLR 2023
>
> [4] From Noisy Prediction to True Label: Noisy Prediction Calibration via Generative Model, ICML 2022
>
> [5] Dividemix: Learning with noisy labels as semi-supervised learning, ICLR 2020

---

> > ### Comment · Reviewer_rUmF · 2023-08-15
> > **About the rebuttal**
> >
> > Thanks for the response. I have read over the rebuttal.
> >
> > However, I feel that as significantly simplifying the assumption of the noise label distribution to be uniform is way too strong and limits the usage of the results of the paper. Essentially, all (most) of the experimental results are based on this strong and unrealistic assumption which makes the work less interesting. As a matter of fact, the accurate modeling of the noise label distribution is most difficult part of the problem which I expect to be resolved or partially resolved.
> >
> > Given the fact that a more realistic noise label distribution has been proposed "From Noisy Prediction to True Label: Noisy Prediction Calibration via Generative Model, ICML 2022" with multinomial distribution. The uniform distribution is only a special case of multinomial distribution. I really donot understand that why we need to do another analysis with the special case presented here.
> >
> > Thus, I maintain my original score.
> >
> > One reviewer.

---

> > > ### Author Response · Authors · 2023-08-15
> > > **We did not assume noisy label distribution is uniform**
> > >
> > > We believe there are misunderstandings of our paper, thus we need to clarify that: **We assume noisy label distribution is multinomial, just like NPC did**.
> > >
> > > - The prior label $p_i(j)$ we constructed contains multiple labels (Eq. 10). The second term **label coverage $c_i(j)$** is where we model noisy label distribution and expect it matches clean label distribution, which is **a multinomial distribution sampled once (Categorical)** (Eq. 11 and Line 183). Furthermore, we prove in ablation Fig. 3, clean label coverage keeps increasing over the training. If we really assume noisy label distribution is uniform, this metric would be fix value.
> > >
> > > - The third term **label uncertainty $u_i(j)$** is indeed sampled from uniform distribution. However, the role of this term is never model noisy  label distribution but **compensate how likely the sample is noisy-labelled**. As we showed in Eq. 10, Z is a normalisation factor that makes $p_i(j)$ sum to 1. The more likely $\tilde{y}_i$ is noisy label ($w_i$ close to 1), the more labels $u_i(j)$ will be sampled and the more uniform $p_i(j)$ becomes. This results in less confidence in prior label compared with one-hot labels and reduce the fitting speed for noisy labelled samples.
> > >
> > > - As shown in *Decompositional Generation Process for Instance-Dependent Partial Label Learning, ICLR 2022*, the prior label we constructed is reasonable in partial label learning and justify for noisy label learning.
> > >
> > > There are many other questions proposed by Reviewer rUmF original comment. Does our rebuttal address reviewer's concerns?

---

### Official Review · Reviewer_6r3h · 2023-07-06

**Soundness:** 3 good
**Presentation:** 4 excellent
**Contribution:** 3 good
**Rating:** 7
**Confidence:** 4

**Summary:**

This paper focuses on improving the efficiency of the generative model in the context of learning noisy labels. To achieve this, the authors first introduce a generative framework whose loglikelihood given a variational posterior can be extended into a label transition term and two KL-divergence terms. Then, the authors demonstrate that the optima of one of the KL-divergence terms could be used to transform a discriminative model into an implicit generative model without extra computation cost.

To further improve the performance of the proposed framework, the authors also present an informative label prior which combines benefits from both the high coverage (sample from Categorical distribution) and the low uncertainty (sample from Uniform distribution) as well as the information contains in the noisy labels.

The proposed work has two main contributions: 1) derive a KL-divergence term from the generative model that allows a discriminative model to be transformed into an implicit generative model, which guarantees the performance of a generative model and the efficiency of a discriminative model at the same time for learning noisy labels; 2) propose a novel clean label prior which allows a tradeoff between the label coverage and uncertainty. Although the second contribution is not as novel in terms of its simplicity, the visualization of its coverage and uncertainty enhances its impact.

The results of the wide range of experiments on benchmark datasets for noisy labels demonstrate the effectiveness of the proposed framework. The ablation studies are also carefully designed and conducted to validate the contribution of each component of the framework.

###########################################################################

######################### Post Rebuttal ######################################

###########################################################################

The authors have addressed all of my concerns. I am happy to raise my score from 6 to 7.



**Strengths:**

The paper is well-presented and easy to follow, with detailed descriptions of each component. The ablation of the framework and the analysis for the proposed clean prior are well performed. The performance gains on CIFAR benchmarks are significant, especially when the instance-dependent noise ratio is high. This indicates that the proposed model could indeed improve the model's robustness against noisy labels.

Minimizing the KL-divergence term KLD( q(y|x) || p(x|y)p(y) ) to estimate the generative model parameters using discriminative model parameters is novel. In addition, the contribution regarding the clean label prior makes this paper a valuable addition to the community.


**Weaknesses:**

Minor:

1. The authors should elaborate more on how the modeling P(X|Y) contributes to the informativeness of noisy labels in the introduction part.

2. Please check the references. Some of them are incomplete (e.g., no journal or conference name for [37]).

**Questions:**

In the related work, the authors mention that the estimation of the label transition matrix is troublesome. Based on equation (7), the estimation of the label transition matrix is still required for the proposed framework. I think the authors did not mention how their optimization of the label transition matrix (the cross-entropy term) differed from the existing work. So, how does the proposed framework alleviate the problems faced by label transition estimation?

---

> ### Author Rebuttal · Authors · 2023-08-03
>
> We thank Reviewer 6r3h for the insightful comments.
>
> > Modeling P(X|Y) contributes to the informativeness of noisy labels.
>
> As shown in Eq.5, $P(X|Y)$ is modelled with the variational posterior $q(Y|X)$. In turn, $q(Y|X)$ depends on the modelling of the latent clean label $Y$. Inspired by partial label learning, we represent $Y$ with a partial label distribution $P(Y)$ that can be seamlessly integrated as supervisory signal without significant modify the framework. Motivated by [1], we define the informativeness of $P(Y)$ as a clean-label coverage of the true label and a regularisation for the complementary labels (i.e., all other labels)
>
> > References incomplete.
>
> We will fix this PiCO[2].
>
> > how does the proposed framework alleviate the problems faced by label transition estimation?
>
> The main issue faced by noisy-label learning methods that rely on label transition estimation is the identifiability of the transition matrix, which forces researchers to make compromising assumptions, such as masking, separability, rankability, clusterability, or the existence of anchor points. Recall that our causal generative process in Fig. 1 allows the factorisation of the joint distribution $P(\tilde{Y},X,Y)$ as $P(\tilde{Y},X,Y) = P(Y)P(X|Y)P(\tilde{Y}|Y,X)$. The dependence on $P(X|Y)$ in the equation above will reduce the uncertainty of the distribution $P(\tilde{Y}|Y,X)$ and encourage the identifiability of the transition relationship, as indicated in CausalNL[3] without making any compromising assumption.
>
> ***
> [1] Decompositional Generation Process for Instance-Dependent Partial Label Learning, ICLR 2023
>
> [2] Pico: Contrastive label disambiguation for robust partial label learning, ICLR 2022
>
> [3] Instance-dependent Label-noise Learning under a Structural Causal Model, NeurIPS 2021

---

> > ### Comment · Reviewer_6r3h · 2023-08-17
> > **About the Rebuttal**
> >
> > I would like to thank the author for the detailed response to my concerns.
> >
> > Now I understand how the proposed causal generative process alleviate the problems face by label transition estimation. Modelling the joint distribution by factorizing it with the dependence on P(X|Y) can indeed help to identify the transition relationship.
> >
> > However, the author did not address my concern regarding how modelling P(X|Y) improve the informativeness of noisy labels. I understand that modelling P(X|Y) will improve the robustness of the model against the noisy label but how such process affect the generation of the noisy label?
> >
> > Reviewer 6r3h

---

> > > ### Author Response · Authors · 2023-08-17
> > >
> > > We thanks for Reviewer 6r3h reply.
> > >
> > > > Transition matrix estimation
> > >
> > > We are pleased that reviewer recognized our argument. The explanation will be updated in manuscript.
> > >
> > > > Informativeness of noisy label
> > >
> > > We feel like there are some confusing points. The informativeness we mentioned in the introduction part is with *clean label prior* $P(Y)$ (Line 51, 58, 76-78), whether it is non-informative (uniform distribution as previous methods used) or informative (our methods). But it is not with noisy label.
> > >
> > > > $P(X|Y)$ and generation of noisy label
> > >
> > > The generation of noisy label is assumed to be instance-dependent (Fig. 2), under the joint effect of $X$ and $Y$. Without modeling $P(X|Y)$, $P(X)$ and $P(Y)$ need to be estimated separately and $P(X)$ is certainly a generative term which requires further decomposition (like latent variable $Z$ for common generative task).
> > >
> > > With the help of modeling $P(X|Y)$ directly, the noisy label generation only requires estimating $P(Y)$ and $P(X)$ can be approximated by $P(Y)$. In other words, $P(X|Y)$ reduces the instance-dependent noise modeling from double variables to single variables, which significantly save the computation cost.

---

> > > > ### Comment · Reviewer_6r3h · 2023-08-18
> > > > **Disscussion**
> > > >
> > > > I greatly appreciate the authors' detailed elaboration.
> > > >
> > > > Initially, I thought that the term "informativeness" in the paper referred to the informativeness of the noisy label. Now, I have a comprehensive understanding of the proposed work.
> > > >
> > > > One more suggestion for the authors is to consider enhancing the clarity of certain terms, such as "informativeness," in the camera-ready version. These terms are widely employed in the community, yet various sub-areas utilize them to denote distinct concepts.
> > > >
> > > > The authors have thoroughly addressed all of my concerns. Consequently, I am happy to raise my score from 6 to 7.
> > > >
> > > > Hope everything goes well when discussing with other reviewers :)
> > > >
> > > > Best,
> > > >
> > > > Reviewer 6r3h

---

> > > > > ### Author Response · Authors · 2023-08-18
> > > > >
> > > > > We really appreciate author's recognition of our work.
> > > > >
> > > > > Indeed, we realized "informativeness" is also used in active learning and uncertainty estimation with different meanings. We will change with a more suitable term.
> > > > >
> > > > > Sincerely,
> > > > >
> > > > > Author

---

### Official Review · Reviewer_9Aeh · 2023-07-07

**Soundness:** 3 good
**Presentation:** 2 fair
**Contribution:** 3 good
**Rating:** 5
**Confidence:** 5

**Summary:**

Most previous works address learning with noisy labels with discriminative models while this paper takes the generative approach which maximizes directly on associated data and clean labels. This generative model is implicitly estimated with a discriminative model, making it computationally more efficient.

**Strengths:**

- This paper successfully addresses several issues that might exist for approaches rely on generative models e.g. challenging to train and tend to use uninformative clean label priors.

- The experiments are extensive with results on many datasets with both realistic and synthetic label noise. All results show the effectiveness of the proposed method.

**Weaknesses:**

- The method is not well motivated. Even though most previous methods often adopt discriminative models and generative models are less discussed, it is still very hard for the reviewer to understand why generative is better than discriminative models for noisy label problems.

- The author argues that the small loss hypothesis offers little guarantee of successfully selecting clean-label samples, however, this hypothesis is very related to the early-learning phenomenon in which the clean labels tend to fit earlier in the training than the noisy labels. This paper still uses this early learning to estimate the clean label prior.

- Figure 2 is difficult to read and it does not make understanding the method any easier, the presentation of the paper should be significantly improved.

**Questions:**

- How does this generative method deal with the case when input $X$ is very large?

- Is there any intuition why the reversed KL loss eventually produces better results?


**Limitations:**

The authors adequately addressed the limitations.

---

> ### Author Rebuttal · Authors · 2023-08-03
>
> We thank Reviewer 9Aeh for the insightful comments.
>
> > Hard for the reviewer to understand why generative is better than discriminative models for noisy label problems.
>
> It is still unclear which method is more suitable for noisy label learning, generative or discriminative. In fact, generative method is less discussed because it requires auxiliary generative module compared with discriminative, as noted by Reviewer 9Aeh. However, as discussed in CausalNL [1], generative models encourages the identifiability of the transition matrix by optimising $p(X |Y )$. In addition, the generative  formulation does not require any additional regularisation [2,3,4] for transition matrix estimation. Our method enjoys the advantages from both sides because we optimize in generative goal but train in discriminative approach. Nevertheless, we believe that the debate of which method (generative or discriminative) is more appropriate to noisy-label learning still needs further study.
>
> > This paper still uses early learning to estimate the clean label prior.
>
> We agree, but it is important to emphasis that we do not use $w_i$ to select clean-label samples, like previously published papers. Instead, $w_i$ works as a weight to the label prior uncertainty regularisation, where low $w_i$ implies higher uncertainty, producing a more uniform $p(y)$ and a slower training convergence compared to a high $w_i$ that produces a $p(y)$ closer to a one-hot distribution, which leads to a faster training convergence.
>
> > Figure 2 is difficult to read.
>
> We have updated Figure 2 in the attached PDF.
>
> > input  $X$ is very large?
>
> The image resolution of $X$ does not have a significant impact on our framework since we approximate the generative model with a discriminative approach, as explained in Eq. 8. Hence, we do not actually generate images during training. The only impact that a large $X$ could have in our model would be in terms of the size of the mini batch for training, which is the same impact as for other discriminative models.
>
> > Reversed KL works better?
>
> The reverse $\mathsf{KL} \left[c_i\times \frac{g_{\theta}(\mathbf{x_i})}
> {\sum_j g_{\theta}(\mathbf{x_j})}\odot \mathbf{p_i}   \Big \|  g_{\theta}(\mathbf{x_i}) \right]$
> can be decomposed into the negative entropy of the first term plus the cross entropy between the first and second terms.
> So, by minimising this loss, we aim to maximise the entropy of the first term and  the cross entropy between the first and second terms.
> For samples where the training is certain about their training labels, with $\tilde{\mathbf{y_i}} \approx \mathbf{c_i}$ and $w_i$ close to 1 , $\mathbf{p_i}$ will be close to a one-hot label.
> In this case, the reverse KL will have a constant entropy value for the first term and a standard cross-entropy loss for the second term.
> On the other hand, the original KL loss
> always maximises the entropy of $g_{\theta}(\mathbf{x_i})$, and the cross entropy between $g_{\theta}(\mathbf{x_i})$ and $c_i\times \frac{ g_{\theta}(\mathbf{x_i})}{\sum_j g_{\theta}(\mathbf{x_j})} \odot \mathbf{p_i}$ independently if the training is certain about the sample's training label.
> The subtle difference of the reverse KL loss leads to a better gradient landscape and better training results.
>
>
>
>
> ***
> [1] Instance-dependent Label-noise Learning under a Structural Causal Model, NeurIPS 2021
>
> [2] Instance-dependent label-noise learning with manifold-regularized transition matrix estimation, CVPR 2023
>
> [3] Provably end-to-end label-noise learning without anchor points, ICML 2022
>
> [4] Estimating instance-dependent bayes-label transition matrix using a deep neural network, ICML 2022

---

> > ### Comment · Reviewer_9Aeh · 2023-08-16
> >
> > Thanks for your response, and thank you for answering my questions.
> >
> > I feel the weak points of the paper mainly stem from the motivations:
> >
> > First, the motivation for proposing a generative model is not sufficient, since as authors agreed that there is no obvious advantage of the generative model over the previous discriminative model. And in order to get this generative model to work, assumptions are made, and tricks are included, making the method looks complex.
> >
> > Second, part of the motivation is based on previous methods that rely on the small-loss hypothesis to select samples, while this paper uses early learning to estimate clean labels, from the reviewer's point of view, these two methods are fundamentally based on similar assumptions -- the clean labels are learned first.
> >
> > Based on these weaknesses, I will keep my original score.

---

> > > ### Author Response · Authors · 2023-08-16
> > >
> > > > Advantage of generative modeling and our framework.
> > >
> > > It is hard to find evidence with limited literatures on generative noisy label learning (CausalNL/InstacenGM and NPC are the only related works we aware). These methods prompt identifiability of transition matrix estimation naturally by optimizing $P(X|Y)$, as shown in CausalNL. But the burden is extra generative module (VAE, GAN).
> > >
> > > Similar discussion has been made in other research areas, include semantic segmentation [1], uncertainty estimation [2] and OOD detection [3]. We believe the advantages of generative modeling is still under exploration and our framework provides a less expensive way to achieve this goal.
> > >
> > > > Our method is complex.
> > >
> > > Our framework derives from Bayes rule (Eq. 4-7) with one straightforward assumption. It builds a plausible approach for optimizing generative model with no extra cost. Furthermore, our method unifies noisy label learning, generative optimization and partial label learning in a single framework (as discussed with Reviewer jVwM). We achieve competitive result both in performance and efficiency. Thus, we respectfully believe our method is not complex.
> > >
> > > > Early learning to estimate clean labels
> > >
> > > We agree with reviewer on this point. However, we also showed results by *not using early learning* in Supplementary file Tab. 2 label coverage with different $\beta$. By setting $\beta=0$ or $\beta=0.5$, we excluded early-learning in our framework and still achieved reasonable performance.
> > >
> > > Anyway, we thanks for Reviewer 9Aeh valuable comments. We will update our paper with better motivation claim.
> > >
> > > ---
> > > [1] GMMSeg: Gaussian Mixture based Generative Semantic Segmentation Models, NeurIPS 2022
> > >
> > > [2] Generative classifiers as a basis for trustworthy image classification, CVPR 2021
> > >
> > > [3] Input complexity and out-of-distribution detection with likelihood-based generative models, ICLR 2019

---

### Author Rebuttal · Authors · 2023-08-08


- **Reviewer 9Aeh, Fig.2 is difficult to read**. We have uploaded a new figure to describe our proposed framework more clearly.

- **Reviewer jVwM, result from other kinds of dataset**. we have uploaded new results from two public NLP news topic classification benchmarks. The baselines are selected from "DyGen: Learning from Noisy Labels via Dynamics-Enhanced Generative Modeling, KDD 2023".

---

### Decision · Program_Chairs · 2023-09-21

**Decision:**

Reject

**Comment:**

This paper proposes a Bayesian framework for learning from noisy labels, which formulates the problem from a generative perspective. Experiments show better results on several baseline datasets compared to some related methods.

While most reviewers gave positive scores, one reviewer thinks the paper needs significant improvement and only gave a score of 1. After rebuttal, the reviewer did not change his mind. I have checked the paper, rebuttal and reviews, while I agree the method is somewhat interesting, I also agree with some points raised by the reviewers, which are not sufficiently addressed, e.g.,

1. The motivation of the proposed generative model for solving the noisy label problem needs to be improved, especially on the advantage compared to existing methods.
2. More experiments on real datasets are needed to further demonstrate the usefulness of the proposed method. From a personal experience, I think the reported results are not state of the art on Clothing1M, and there are more real datasets including ImageNet, Food-101N etc that need to be evaluated on.
3. The simplified assumptions mentioned by the reviewer seem OK for me to be used as priors, however, I encourage the authors to clarify and emphazie the reasonability in the revision.